# Case Report of Infectious Spondylitis Caused by *Kytococcus sedentarius*

**DOI:** 10.3390/medicina57080797

**Published:** 2021-08-02

**Authors:** Kyoung Ree Lim, Jun Seong Son, Soo-youn Moon

**Affiliations:** 1Department of Medicine, College of Medicine, Kangwon National University, Chuncheon 24289, Korea; idkrlim@naver.com; 2Division of Infectious Diseases, Department of Internal Medicine, Kyung Hee University Hospital at Gangdong, Seoul 05278, Korea; isonjs@naver.com

**Keywords:** *Kytococcus sedentarius*, infectious spondylitis, *Kytococcus* spp., spine infection

## Abstract

Background: *Kytococcus* species has not been considered a pathogen, but infections caused by this species are increasing. There are several cases of infections caused by *Kytococcus sedentarius,* but no case of infectious spondylitis has been reported yet. Case presentation: A 79-year-old female patient was hospitalized because of back pain for several months. She was diagnosed with infectious spondylitis, and *K. sedentarius* was cultured from the pus and specimen obtained during the surgical procedure. The patient recovered completely without recurrence after 6 months of treatment with ciprofloxacin alone for 8 weeks. Conclusion: This is the first case report of infectious spondylitis caused by *K. sedentarius*.

## 1. Introduction

*K. sedentarius* is generally considered a harmless commensal, and infection caused by this organism is not common [1,2]. *Kytococci* are gram-positive, pigmented, non-encapsulated, non-motile, aerobic, catalase-positive cocci appearing in pairs or tetrads. The genus consists of three species: *K. sedentarius, K. schroeteri*, and *K. aerolatus*. [3]

Infectious spondylitis is an infectious disease of the bony spinal column and the intervertebral discs, which is increasing in the aging population [4]. Magnetic resonance imaging (MRI) and computed tomography (CT) are good tools in diagnosing infectious spondylitis and detecting complications such as an abscess [5].

To our knowledge, there have been no reports of spondylitis caused by *K. sedentarius*. Here, we present a case of postoperative spondylitis due to *K. sedentarius.*

This case report was reviewed by the institutional review board of the hospital. As we could not contact the patient or any family members after treatment, the informed consent was waivered by IRB.

## 2. Case Presentation

A 79-year-old woman was admitted to a hospital for increasing back pain and left leg radiating pain for several months. She had type 2 diabetes mellitus and hypertension and was taking medications. She was treated with physiotherapy and injection for her back and radiating pain every other week for several months before her back pain was exacerbated.

On admission, her vital signs were stable, with a blood pressure of 138/75 mmHg, pulse rate of 75/min, respiration rate of 20/min, and body temperature of 36.7 °C. Blood tests on admission showed an elevated erythrocyte sedimentation rate (ESR) of 73 mm/h and C-reactive protein (CRP) level of 6.0 mg/dL. ESR and CRP are the most common tests to diagnose and monitor inflammatory conditions [6]. MRI showed infectious spondylitis with an abscess at the posteroinferior aspect of the L5 body and epidural enhancement at the L4-S2 level. (Figure 1a)

A two-stage operation was planned. The first surgery was performed on the 3rd day of hospitalization. Laminectomy and facetectomy were performed at the L5-S1 spine with massive irrigation. After surgical debridement and collection of culture specimens, empirical antimicrobial therapy with cefotiam was started before the culture results became available. Blood culture was performed before starting empirical antimicrobial treatment, in which no micro-organism was identified *Kytococcus sedentarius* was identified from a closed pus and tissue culture obtained during surgery. A VITEK^®^ II system (bioMérieux, La Balme-Les-Grottes, France) and matrix-assisted laser desorption ionization-time of flight mass spectrometry (MALDI-TOF) (bioMerieux, Marcy-l’Etoile, France) were used to identify the causative pathogens. The antibiotic susceptibility test showed minimum inhibitory concentrations (MICs) of ≤0.5 mg/L for ciprofloxacin, 16 for vancomycin, ≥4 mg/L for oxacillin, and ≥32 mg/L for rifampin. Once the antimicrobial sensitivity test results became available after 14 days, the antimicrobial therapy was changed to ciprofloxacin. On the 15th day of hospitalization, a second operation was performed to secure the spine’s stability by posterior lumbar interbody fusion.

Intravenous ciprofloxacin was continued for 2 weeks before hospital discharge and switched to oral ciprofloxacin, which was continued for 6 more weeks.

After a total of 8 weeks’ treatment, follow-up MRI showed no fluid collection, abscess, or epidural enhancement (Figure 1b). Lab tests showed an ESR of 26 mm/h and a CRP concentration of 0.3 mg/dL.

The 6-month follow-up at the outpatient clinic showed that the patient’s back pain and radiating leg pain had been alleviated. The blood tests showed an ESR of 26 mm/h and CRP concentration of 0.2 mg/dL.

## 3. Discussion

The significance of *Kytococcus* as a human pathogen may not have been fully recognized, or infections were previously misidentified as *Micrococcus* spp. [7] *Kytococcus* was separated from *Micrococcus* based on phylogenetic and chemotaxonomic analyses, with *K. sedentarius* the first species described in the genus *Kytococcus* in 1995 [8]. The recent identification of *Kytococcus* infection has been enabled by molecular sequencing or MALDI-TOF MS. An increased number of infections caused by *K. schroeteri* has been reported recently [7].

According to Shah’s study on *K. schroeteri* infections, 13 of 19 cases were related to prostheses, while the others were pneumonia cases [7]. Only one case of postoperative spondylitis was reported in a female patient with diabetes. Among the 19 patients, 11 were immunocompromised.

*K. sedentarius* is usually a skin organism that is not harmful but can be associated with infections [1,2,9]. Few *K. sedentarius* infections have been reported, including pneumonia in a patient with acute leukemia and two cases related to prostheses (Table 1). As *K. sedentarius* is normal skin commensal, breakage of the skin barrier due to prosthetics or injection therapy could lead to infection. To our knowledge, this is the first case of infectious spondylitis caused by *K. sedentarius.*

Regarding the treatment of this strain, there are no formally established MIC breakpoints from the Clinical Laboratory Standards Institute (CLSI) for *Kytococcus* species. Therefore, the MIC breakpoints should be interpreted with caution. *Kytococcus* species are usually resistant to methicillin and penicillin G [10] but susceptible to streptomycin, novobiocin, tetracycline, vancomycin, and polymyxin B [8]. Although there are no data on in vitro susceptibility test results of fluoroquinolones for *Kytococcus* spp., there are data on the in vitro activity of antibiotics for *Micrococcus* spp. [11,12]. According to Peters, levofloxacin and ciprofloxacin were equally active against all 191 micrococcal strains tested and twofold more active than ofloxacin [11].

In the present case, as the isolate was resistant to most antimicrobial agents, including vancomycin and rifampin, the patient was treated with ciprofloxacin alone. Two reported cases of *K. schroeteri* infections were treated with fluoroquinolones. One patient with pneumonia died with ofloxacin and ceftriaxone treatment; the other was a case of prosthetic discitis that survived with ofloxacin and rifampin treatment [7]. The difference between this case and other *K. schroeteri* cases was that we treated the patient with ciprofloxacin alone, which is considered twice as effective as ofloxacin [11].

## 4. Conclusions

In summary, we described the first known case of infectious spondylitis caused *by K. sedentarius*, which was treated with surgery and an antibiotic agent. Although *Kytococcus* species are considered part of the skin’s normal flora, human infections with *K. sedentarius* do occur, especially if they are related to invasive procedures. When culture studies show *K. sedentarius* in patients with infectious spondylitis, it should be considered a pathogen rather than a contaminant. In the treatment of *K. sedentarius* infection, ciprofloxacin was a successful treatment for infectious spondylitis, although more cases are needed.

## Figures and Tables

**Figure 1 medicina-57-00797-f001:**
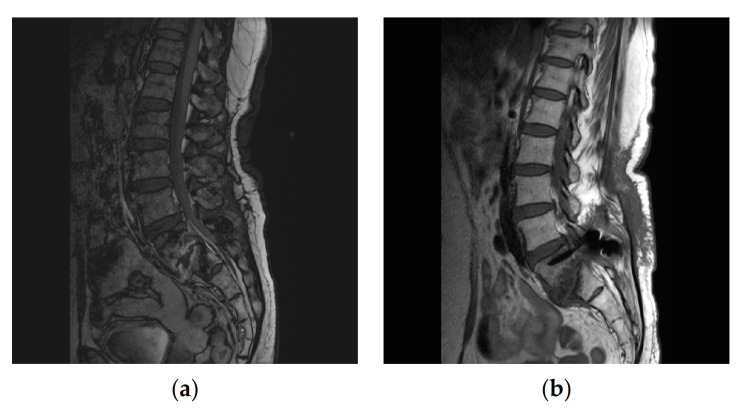
Spine magnetic resonance imaging (MRI) showing infectious spondylitis of L5-S1 with epidural enhancement at the L4-S1 level with decreased disc height and bone marrow signal change before treatment (**a**) and improvement of fluid collection with no newly developed bony destructive change at and around the L5-S1 level (**b**).

**Table 1 medicina-57-00797-t001:** Reported cases of *Kytococcus sedentarius* infection.

Sex/Age (Years)	Immunocompromising Condition	Primary Source of Infection	Therapy	Outcome	Report
F/66	End-stage renal disease due to type 2 diabetes mellitus	Peritoneal dialysis-associated peritonitis	Vancomycin intraperitoneal injection	Recovered	Chaudhary et al. [1]
M/55	Acute myeloid leukemia	Hemorrhagic pneumonia	Teicoplanin and cefepime	Deceased	Levanga et al. [2]
M/67	None	Femoropopliteal bypass graft infection	Amoxicillin-clavulanate	Recovered	Dainese et al. [6]

## Data Availability

Data is contained within the article.

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
