# Peer review of "Case Report of Infectious Spondylitis Caused by Kytococcus sedentarius"

_medicina, 2021, doi:10.3390/medicina57080797_

Round 1

Reviewer 1 Report

  1. Sub-par writing quality, including scattered spelling errors (even in the abstract)
  2. The authors did not discuss the possibility that the infection might have developed independently, and not necessarily "caused" the spondylitis. 
  3. Infectious spondylitis is quite well explained in this article (DOI: 10.1007/s001170050142).
  4. From the previous article, it might seem that this work is not as novel as authors claim. The authors need to address how their paper adds/compares to the previous work.
  5. The elevated ESR is indicative of inflammation (DOI: 10.1182/blood-2018-99-117260). Authors should include this information, since all the readers may not be familiar with the measure, and fail to see the relevance.

Author Response

1. Sub-par writing quality, including scattered spelling errors (even in the abstract)

→ The manuscript was sent for English editing service. Those spelling errors were checked again. If it is necessary, we will ask for another English editing service.

2. The authors did not discuss the possibility that the infection might have developed independently, and not necessarily "caused" the spondylitis.

→ K. sedentarius was cultured from the closed pus and specimens from the site of infectious spondylitis, obtained during the surgery. So it is considered a causative organism.

3. Infectious spondylitis is quite well explained in this article (DOI: 10.1007/s001170050142).

→  We checked the article about infectious spondylitis in adults. The article is mainly focused on the imaging diagnosis of infectious spondylitis. We added comments on infectious spondylitis in introduction.

4. From the previous article, it might seem that this work is not as novel as authors claim. The authors need to address how their paper adds/compares to the previous work.

→ There were several case reports of infectious spondylitis by K. schroeteri, but no case of infectious spondylitis by K. sedentarius yet. Although there were some cases of K. sedentarius, none of them were infectious spondylitis. That makes this case different from previous case reports.

5. The elevated ESR is indicative of inflammation (DOI: 10.1182/blood-2018-99-117260). Authors should include this information, since all the readers may not be familiar with the measure, and fail to see the relevance.

→ I added a comment on the meaning of ESR and CRP in the case presentation section.

Reviewer 2 Report

The authors present a case of vertebral osteomyelitis related to the bacteria K. sedentarius. The article is concise and clear. I have minor observations:

-Line 9: correct spelling and italize Kytococcus.

-Line 10: increasing, or increasingly reported?

-Line 25: I don't think the term postoperative seems the most adequate term, maybe postprocedural or something similar would be better.

-I would like to know if blood cultures were performed and the size of the abscess as per MRI, if available.

-It would be interesting to know why the patient was treated surgically instead of conservatively. I believe this is mainly a center-dependent approach, but maybe other data may help explaining why the team decided on surgical treatment. Consider adding a minor comment if appropiate. I think in my hospital, unless the size of the abscess was very large, the patient would have been treated conservatively, with antibiotic treatment guided by thick needle aspirate culture.

-References: please unify style, specially with the journals' names (differences in capitalizing and abbreviating)

Author Response

-Line 9: correct spelling and italize Kytococcus.

→ The errors were corrected as mentioned.

-Line 10: increasing, or increasingly reported?

→ I rewrite the sentence to 'increasingly reported' to make it more clear.

-Line 25: I don't think the term postoperative seems the most adequate term, maybe postprocedural or something similar would be better.

→ I have changed the word ‘postoperative’ to ‘post-procedural.'

-I would like to know if blood cultures were performed and the size of the abscess as per MRI, if available.

→ The sized of the abscess was 1.5 cm x 2.1 cm, which is mentioned in case report. Blood culture was performed before antimicrobial therapy, which is also added.

-It would be interesting to know why the patient was treated surgically instead of conservatively. I believe this is mainly a center-dependent approach, but maybe other data may help explaining why the team decided on surgical treatment. Consider adding a minor comment if appropiate. I think in my hospital, unless the size of the abscess was very large, the patient would have been treated conservatively, with antibiotic treatment guided by thick needle aspirate culture.

→ The patient had increasing pain, for which surgical treatment was an option. Also the pathogen can be identified by obtaining surgical specimen. I mentioned it in the manuscript.

-References: please unify style, specially with the journals' names (differences in capitalizing and abbreviating)

→ I unify reference style.

Round 2

Reviewer 1 Report

The authors have satisfactorily responded to the questions.